# HSF1 Stimulates Glutamine Transport by Super-Enhancer-Driven lncRNA LINC00857 in Colorectal Cancer

**DOI:** 10.3390/cancers14163855

**Published:** 2022-08-09

**Authors:** Qi Shen, Rui Wang, Xinling Liu, Ping Song, Mingzhu Zheng, Xiaomin Ren, Jingang Ma, Zhong Lu, Jiaqiu Li

**Affiliations:** 1Department of Radiation Oncology, The First Affiliated Hospital of USTC, Division of Life Sciences and Medicine, University of Science and Technology of China, Hefei 230002, China; 2Department of Oncology, Clinical Research Center, Affiliated Hospital of Weifang Medical University, Weifang 261031, China; 3Department of Gastroenterology, Affiliated Hangzhou First People’s Hospital, Medical School of Zhejiang University, Hangzhou 310006, China

**Keywords:** colorectal cancer, HSF1, super enhancer, glutamine metabolism, LINC00857

## Abstract

**Simple Summary:**

Based on the latest research, cancer cells prefer glutamine to glucose. Therefore, it is more worthwhile to explore the regulatory mechanism of glutamine metabolism in cancer cells. Super enhancers are critical for the gene transcriptional programs responsible for cell fate by interacting with various transcription factors. The transcription factor HSF1 exerts a multifaced role in tumorigenesis. However, the relevance of HSF1 to super enhancers in tumors remains obscure. Therefore, this study focused on the mechanism underlying super-enhancer activation and its relationship to HSF1 in CRC. Here, we performed a super-enhancer landscape in CRC and we screened out an HSF1-mediated super enhancer, lncRNA-LINC00857, by lncRNA microarray. We discovered that HSF1 could stimulate acetyltransferase P300-mediated super-enhancer activity to facilitate LINC00857 expression, contributing to SLC1A5/ASCT2-mediated glutamine transport. In addition, we validated that targeting the HSF1/LINC00857/ANXA11 axis may provide a valuable therapeutic strategy against CRC.

**Abstract:**

Super enhancers are critical for the gene transcription responsible for cell fate by interacting with transcription factors. However, the relevance of HSF1 to super enhancers in tumors remains obscure. We profiled H3K27ac enrichment by chromatin immunoprecipitation sequencing. HSF1-mediated lncRNAs were identified by lncRNA microarray. The characteristics of LINC00857 were explored by in vitro and in vivo assays. The mechanism was studied via chromatin immunoprecipitation, RNA immunoprecipitation, and HSF1/ANXA11 knockout mice. We found that super enhancers occupied multiple gene loci in colorectal cancer. We screened out an HSF1-mediated super enhancer, lncRNA-LINC00857, which exerts its characteristics in promoting cell growth via regulating glutamine metabolism. Notably, HSF1 could stimulate the super-enhancer activity of LINC00857 by the enrichment of acetyltransferase P300 to its gene loci, contributing to LINC00857 transcription. In turn, nuclear LINC00857 cooperated with HSF1 to promote ANXA11 transcription, which modulated SLC1A5/ASCT2 protein expression by binding competitively to miR-122-5p. The knockout of ANXA11 attenuated colorectal cancer formation in vivo. Collectively, we shed light on a closely cooperative machinery between HSF1 and super enhancers. HSF1 could stimulate acetyltransferase P300-mediated super-enhancer activity to facilitate LINC00857 expression, contributing to SLC1A5-mediated glutamine transport. Targeting the HSF1/LINC00857/ANXA11 axis may provide a valuable therapeutic strategy against colorectal cancer.

## 1. Introduction

Colorectal cancer (CRC), a high-incidence malignancy, ranks second in terms of mortality globally [1]. CRC is a multifactorial disease, with factors such as genetic predisposition and dysbiosis of the gut microbiota [2]. Once diagnosed, however, most CRC patients are at an advanced stage, and the recurrence rate of CRC is also high due to the infiltration of surrounding tissues and distant metastasis. Therefore, further exploration of the mechanisms of CRC carcinogenesis is beneficial for diagnosis and treatment.

Super enhancers (SE) are clusters of multiple adjacent enhancers. Specific histone modifications are essential for super-enhancer activity, especially histone 3 lysine 27 acetylation (H3K27ac) which is the most common marker used to identify super enhancers. Unlike typical enhancers, super enhancers have a larger genome region span and most of their associated genes play key roles in cell-fate determination [3,4]. Targeting super enhancer-driven genes offers an Achilles heel for killing cancer cells [5]. Super enhancers exert powerful transcriptional regulation by recruiting a wealth of master transcription factors, RNA polymerase II, and other cofactors [6,7]. The bromodomain and extra-terminal (BET) proteins such as BRD4 serve as primary epigenetic “readers” that activate gene transcription by identifying and binding histone acetylated lysine residues at the region where the super enhancer is located [8,9]. Therefore, super-enhancer-related genes tend to be sensitive to BET inhibitors such as JQ1 and I-BET-762 [10,11]. Super enhancers can also generate noncoding RNA [12]. Most of the super-enhancer noncoding RNA belongs to the class of lncRNA, called super-enhancer lncRNA (SE-lncRNA), which is seldom studied.

As an evolutionarily conserved transcription factor, heat shock factor 1 (HSF1) protect cells from cellular stress responses [13]. Additionally, HSF1 has been found to be overexpressed or activated in a variety of cancers [14]. Recent years have witnessed the prominent roles of HSF1 in processes such as maintaining protein homeostasis, repairing damaged DNA, facilitating cell proliferation, migration, and regulating the process of apoptosis [15,16]. Unexpectedly, HSF1 could affect tumor immunity by regulating MHC-I machinery or PD-L1 expression [17,18]. Notably, HSF1 is closely associated with metabolism reprogramming, which is considered to be one of the hallmarks of cancer [19]. HSF1 has been proven to be intimately involved in glycose, lipid, and amino-acid metabolism [20,21,22]. By regulating tumor metabolic processes, HSF1 can help tumor cells to adapt to adverse environments. The “pioneer” transcription factor HSF1 could first get close to the regulatory elements and help the recruitment of other factors by acting as a scaffold [23,24]. Therefore, this study focused on the mechanism underlying super-enhancer activation and its relationship with HSF1 in CRC. Here, we performed a super-enhancer landscape in CRC, and we screened out an HSF1-mediated super-enhancer, lncRNA-LINC00857, by lncRNA microarray. Importantly, HSF1 could stimulate acetyltransferase P300-mediated super-enhancer activity to facilitate LINC00857 expression, contributing to SLC1A5-mediated glutamine transport. In addition, we validated that targeting the HSF1/LINC00857/ANXA11 axis may provide a valuable therapeutic strategy against CRC.

## 2. Materials and Methods

### 2.1. Cells

All cells were purchased from the cell bank of the Chinese Academy of Sciences (Shanghai, China). HCT116 and DLD1 were cultured at 37 °C with 5% CO_2_ in McCOY’s 5A medium (GENOM, GNM16600, Hangzhou, China) or RPMI 1640 medium (GENOM, GNM31800) supplemented with 10% fetal bovine serum. The cells were tested for mycoplasma contamination every two months.

### 2.2. Antibodies, Reagents, and Kits

The following antibodies were used in this study: a-Tubulin (T5168, Sigma-Aldrich, St. Louis, MO, USA), HSF1 (ab52757, Abcam, Cambridge, UK), Histone H3 (acetyl K27) (ab177178, Abcam), ANXA11 (SC46686, Santa Cruz, CA, USA), p-mTOR (ab109268, Abcam), p-P70S6K (9234S, CST, Danvers, MA, USA), SLC1A5 (ab237704, Abcam), P300 (ab275378, Abcam), Pol II (ab26721, Abcam), BRD4 (A301-985A100, Bethyl Laboratories, Montgomery, TX, USA), β-actin (4970L, CST). The reagents used in this study are listed below: JQ1 (S7110, Selleck, Shanghai, China), I-BET-762 (S7189, Selleck), X-treme GENE™ 9 DNA Transfection Reagent (6365779001, Roche, Basel, Switzerland), Lipofectamine™ RNAiMAX (Invitrogen, Carlsbad, MA, USA, 13778150), UltraSYBR Mixture (CW0957M, cwbiotech, Taizhou, China), AOM (Azoxymethane) (A5486, Sigma-Aldrich), DSS (dextran sulfate sodium salt colitis grade) (160110 (MW36000–50000), MP). The kits used in this study are listed below: Simple ChIP^®^ Enzymatic Chromatin IP Kit (Magnetic Beads) (CST, 9003S), miRNeasy Mini Kit (50) (Qiagen, Duesseldorf, Germany, 217004), miScript II RT Kit (50) (Qiagen, 218161), miScript SYBR Green PCR Kit (200) (Qiagen, 218073), High Capacity cDNA Reverse Transcription Kit (Thermo, Waltham, MA, USA, 4368814), Magna RIP™ RNA Binding Protein Immunoprecipitation Kit (Millipore, Billerica, MA, USA, no. 17–700), Dual-Luciferase Reporter Kit (Promega, Madison, WI, USA, E2920), RIPAb + AGO2 Kit (EMD Millipore, 03-110).

### 2.3. Tissue Samples

All experiments were in accordance with the Helsinki Declaration. This study was approved by The Institute of Research Medical Ethics Committee of the Affiliated Hospital of Weifang Medical University. All patients signed the informed consent forms. The samples were obtained from the Affiliated Hospital of Weifang Medical University. The patients did not receive other treatments before surgery. The samples were collected after surgery and stored at −80 °C immediately.

### 2.4. CCK8 Assays and Colony Formation Assays

A total of 3000 cells per well were seeded in 96-well plates. Cell Counting Kit-8 (C0038-500, Beyotime, Shanghai, China) solution was added to the cells and incubated at 37 °C for 30 min. Subsequently, the optical density value was checked at a wavelength of 450 nm. A total of 1000 cells per well were seeded in 6-well plates and cultured for 11 days. Then, the colonies were stained by 0.1% crystal violet and calculated.

### 2.5. siRNA and shRNA Transfection

A total of 1 × 10^5^ HCT116 or DLD-1 cells were seeded in 6-well plates and transfected with siRNA by RNAiMAX. The cells were infected with the lentiviral-expressing shRNA via 5 µg/mL polybrene. The siRNA/shRNA was purchased from Genepharma (Shanghai, China). All mentioned sequences are listed in Appendix A.

### 2.6. RNA Isolation, Reverse Transcription, and Quantitative PCR

The qPCR assays were performed as reported previously [22]. a-Tubulin was used for the control. The primer sequences for qPCR are listed in Appendix A.

### 2.7. Western Blot and Immunohistochemistry

WB and IHC assays were performed as previously reported [22]. The antibody concentrations for Western blot were as follows: HSF1 (1:1000), H3K27ac (1:10,000), a-Tubulin (1:10,000), GLS1 (1:1000), ANXA11 (1:1000), SLC1A5 (1:5000), β-actin (1:5000), p-mTOR (1:1000), p-P70S6K (1:1000). The antibody concentrations for IHC were as follows: Ki67 (1:200), p-mTOR (1:100), HSF1 (1:200), ANXA11 (1:400). The original Western Blot figures can be found in Appendix A.

### 2.8. Fluorescence In Situ Hybridization (FISH)

Briefly, the sample slides were fixed with formaldehyde. After dewaxing and dehydration, the denatured probe was hybridized overnight at 42 °C. Subsequently, the slides were rapidly washed, and the nuclei were counterstained with DAPI. The nuclei stained by DAPI were blue under ultraviolet excitation, and the expression of LINC00857 was achieved with green light (FAM (488)). The probe against LINC00857 was purchased from Servicebio (Wuhan, China). The sequence of the probe is as follows: 5′-FAM-TTGGGACAGGGTTTGGAACTCTTGCGG-FAM-3′.

### 2.9. Luciferase Activity Assays

EXO5-3xFlag-HSF1, pGL4.10-LINC00857 promoter, pGL4.10-ANXA11 promoter, pMIR-REPORT-ANXA11 (WT or MT), and pMIR-REPORT-SLC1A5 (WT or MT) plasmids were obtained from OBiO technology (Shanghai, China). The plasmids were transfected into 293T cells via the X-treme GENE™ 9 DNA Transfection Reagent. The luciferase activities were measured via a dual-luciferase reporter kit (Promega, E2920).

### 2.10. Chromatin Immunoprecipitation (ChIP) and Sequencing Analysis (ChIP-Seq)

The ChIP assays were performed as reported previously [25]. The purified DNA was analyzed by qPCR. The primer sequences for qPCR are listed in Appendix A. Meanwhile, high-quality purified DNA was sequenced and analyzed on an Illumina NovaSeq 6000 at Kangchen Bio-tech Inc. (Shanghai, China). MACS V1.4.2 software was used for peak detection. Statistically significant ChIP-enriched peaks were identified by comparison of the IP vs. input sample.

### 2.11. lncRNA Microarray Analysis

Total RNA was isolated via the Trizol reagent (Qiagen, 1023537). The lncRNA expression profile was measured by Arraystar LncRNA microarray V5.0 by Kangchen Bio-tech Inc. (Shanghai, China). The data were analyzed using the limma package in R Studio. Differentially expressed lncRNAs after HSF1 knockdown were screened out when the *p* value < 0.05 and |log_2_FC| ≥ 1.0. The following analysis contains elements of gene ontology (GO) analysis, Kyoto encyclopedia of genes and genomes (KEGG) analysis, and gene set enrichment analysis (GSEA). Firstly, the correlation was calculated between each differential lncRNA and all mRNAs, and then GSEA pre-ranked in GSEA software was used to conduct a gene set enrichment analysis.

### 2.12. RNA Immunoprecipitation Assays (RIP)

The RIP assays were performed as reported previously [25]. The antibodies for RIP were used via the RIPAb + AGO2 Kit (EMD Millipore, 03-110). The RNA was isolated for qPCR analysis. The primer sequences for qPCR are listed in Appendix A.

### 2.13. Amino Acid Metabolism Analysis

A total of 1 × 10^7^ cells were collected and flash-frozen in liquid nitrogen. LC–MS was conducted at Bio Novo Gene Co., Ltd. (Suzhou, China). In brief, the cells were added to 300 μL 10% formic acid methanol–H_2_O (1:1 *v*/*v*) solution and 100 mg glass beads. Then, the cells were immersed in liquid nitrogen for rapid freezing for 5 min. The cells were oscillated cells at 50 Hz for 1 min and centrifuged at 12,000 rpm for 5 min at 4 °C. An aliquot of 20 μL of the supernatant was added to a 180 μL 10% formic acid methanol–H_2_O (1:1 *v*/*v*) solution. Then, 100 µL double isotope was added to achieve a concentration of 100 ppb to 100 µL diluted samples. Finally, the supernatant was filtrated through a 0.22 μM membrane. After LC–MS, the concentration of 22 amino acids was analyzed quantitatively.

### 2.14. Xenograft Model

Animal studies were approved by the Institutional Animal Care and Use Committee of Weifang Medical University and NIH guidelines. Ten 5-week-old BABL/c nude mice were divided into two groups. A total of 5 × 10^6^ HCT116 cells transfected with control-shRNA or LINC00857-shRNA were subcutaneously injected into the nude mice, respectively. Tumor volumes were measured every three days. All mice were sacrificed at day 26, and tumors were weighed. The tumors were fixed with neutral formalin for subsequent immunohistochemical experiments.

### 2.15. AOM/DSS Model

Both the 6-week-old normal and ANXA11 knockout (ANXA11^−/−^) C57BL/6J mice were purchased from GemPharmatech Co., Ltd. (Wilmington, DE, USA) The ANXA11 KO mice were produced by the CRISPR/Cas9 strategy. All mice were induced to an azoxymethane (AOM)/dextran sulfate sodium (DSS) model as reported previously [22]. All mice were sacrificed at day 100. The colorectal tissues were dissected, rinsed in PBS, and cut open longitudinally to measure the tumor numbers and sizes.

### 2.16. Statistical Analysis

The online TCGA database used in this study contains GEPIA (http://gepia2.cancer-pku.cn/ (accessed on 12 January 2021 )) [26], TIMER (http://timer.cistrome.org/ (accessed on 9 January 2022)) [27], Starbase (http://starbase.sysu.edu.cn/ (accessed on 21 February 2021)) [28], lnCAR (https://lncar.renlab.org/ (accessed on 6 March 2021)) [29], Kaplan–Meier Plotter (http://kmplot.com/analysis/ (accessed on 5 April 2021)) [30], UCSC (http://genome.ucsc.edu/ (accessed on 18 June 2021)) [31], and ENCODE (https://www.encodeproject.org/ (accessed on 11 October 2021)) [32]. The student’s *t*-test and log-rank (Mantel–Cox) test were performed for statistical significance analysis using GraphPad Prism 7. A *p* value < 0.05 was considered as statistically significant. * *p* < 0.05; ** *p* < 0.01; *** *p* < 0.001; **** *p* < 0.0001.

## 3. Results

### 3.1. H3K27ac Landscape in Colorectal Cancer

H3K27ac is the best marker for indicating super enhancers. To explore whether super enhancers are associated with CRC, we performed ChIP-seq with H3K27ac antibodies in HCT116 cells (GSE198223). A systematic analysis of the H3K27ac landscape in HCT116 cells was conducted and the top 20 associated mRNAs were annotated (Figure 1A,B). Interestingly, the majority of the H3K27ac ChIP-seq peaks were located at intron and intergenic regions (Figure 1C). To gain insight into the features of the super-enhancer-associated genes in CRC, we subjected them to gene ontology (GO) and Kyoto encyclopedia of genes and genomes (KEGG) analyses (Figure 1D–G). Overall, we profiled H3K27ac enrichment by chromatin immunoprecipitation sequencing in CRC.

### 3.2. Screening for HSF1-Associated lncRNAs

The super-enhancer region is more densely occupied by master transcription factors. The transcription factor HSF1 is closely associated with cell metabolism, which is in accordance with super enhancers. Super enhancers can also generate noncoding RNA, called super-enhancer lncRNA (SE-lncRNA), which is seldom studied in colorectal cancer. Therefore, this study put emphasis on the HSF1-associated SE-lncRNA. Firstly, we performed an lncRNA microarray analysis after knockdown of HSF1 with two specific siRNAs in HCT116 cells (GSE198224) (Figure 2A). We identified 883 significantly upregulated lncRNAs that exhibited a more than 2-fold change, while there were 291 downregulated lncRNAs (Figure 2B). Among them, 111 lncRNAs belonged to enhancer-mediated lncRNAs (Figure 2C). Furthermore, 72 enhancer-mediated lncRNAs (65%) had a canonical HSF1 binding site (HSE) in their promoter regions (Figure 2D), suggesting potential bona fide targets of HSF1. Among the 16 downregulated enhancer-mediated lncRNAs, 8 lncRNAs had an HSE in their promoter regions, demonstrating that enhancer-mediated lncRNAs are closely associated with HSF1 in CRC. Indeed, these enhancer-mediated lncRNAs showed significant H3K27ac signals in HCT116 cells during the analysis of datasets downloaded from the ENCODE database (Figure 2E and Appendix A), which was consistent with our own ChIP-seq data (Appendix A). Subsequently, we performed a gene set enrichment analysis (GSEA) on these eight enhancer-mediated lncRNAs (Appendix A). Unexpectedly, we discovered that LINC00857, which had the highest H3K27ac signal (Figure 2E), showed a distinct GO and KEGG from others. Collectively, our study screened out a series of HSF1-associated lncRNAs, and we selected LINC00857 as the target for the following study.

### 3.3. HSF1 Activates the Transcription of LINC00857 via Promoting Its Super-Enhancer Activity

We discovered clusters of multiple adjacent enhancers in LINC00857 gene loci in CRC cell lines (Figure 3A), suggesting that LINC00857 may be a super-enhancer lncRNA. To further validate this hypothesis, we identified canonical HSF1 binding sites at the LINC00857 promoter and two enhancers (E1 and E2). ChIP-qPCR assays confirmed the occupancy of H3K27ac and HSF1 at these sites (Figure 3B–E). We further constructed a luciferase reporter containing an HSF1 binding site at the LINC00857 promoter that showed a higher H3K27ac activity and found that HSF1 could stimulate the activity of the LINC00857 promoter (Appendix A), suggesting that HSF1 was responsible for LINC00857 promoter activity. Subsequently, we explored whether and how LINC00857 is regulated by HSF1 and super enhancers. The cancer genome atlas (TCGA) data showed a significant positive correlation between the expression of HSF1 and LINC00857 (Appendix A). Additionally, we confirmed that the LINC00857 level was clearly reduced after HSF1 knockdown (Appendix A). BRD4, the most well-studied BET protein, was overexpressed and correlated with the expression of LINC00857 in CRC (Appendix A). Upon the genetic deletion of BRD4 or the inhibition of the BET domain by inhibitors (JQ1 and I-BET-762), which can strongly interfere with the establishment of super enhancers [10], LINC00857 expression and the cell viability was both decreased significantly (Appendix A). These findings indicated that LINC00857 expression was indeed subject to HSF1 and super-enhancer activity. Unexpectedly, we discovered that the level of total H3K27ac activity was significantly reduced after silencing HSF1 (Figure 3F). As expected, the enrichment of H3K27ac at LINC00857 gene loci was decreased greatly after HSF1 knockdown (Figure 3G), demonstrating that the super-enhancer activity of LINC00857 depended on HSF1 expression. The latest research has reported that the acetyltransferase P300 was responsible for the super-enhancer activity [33]. Coincidentally, we observed that the enrichment of P300 at LINC00857 gene loci was markedly increased, while it appeared to reduce after HSF1 knockdown (Figure 3H,I and Appendix A)—implying that HSF1 is indispensable for P300-mediated super-enhancer activity. Consequently, we concluded that HSF1 could activate the transcription of LINC00857 via regulating its super-enhancer activity, which is driven via acetyltransferase P300.

### 3.4. LINC00857 Is Upregulated and Beneficial for CRC Carcinogenesis

Next, we analyzed the roles of LINC00857 in CRC using the data from the TCGA database. As shown in Figure 4A and Appendix A, LINC00857 was upregulated in CRC and correlated with tumor grade and microsatellite instability status. Additionally, CRC patients with a high LINC00857 expression had a shorter overall survival (Appendix A), suggesting its key role in driving CRC progression. Likewise, our FISH assays verified the overexpression of LINC00857 in CRC tissues (Figure 4B). Meanwhile, qPCR assays indicated that LINC00857 was consistently upregulated in CRC cell lines compared with FHC—the normal colon epithelial cell (Figure 4C). CCK8 and colony formation assays implied that the proliferative ability of HCT116 and DLD1 cells dropped dramatically after LINC00857 knockdown by two specific siRNAs (Figure 4D–F). To validate the effect during in vitro assays, we packaged the most effective siRNA sequence into a lentivirus to knock down LINC00857, which showed a similar effect on cell proliferation (Appendix A). In vivo, we conducted xenograft models using HCT116 cells, and we observed that the tumor volume and weight were smaller after LINC00857 knockdown by lentiviral-expressing shRNA (Figure 4G–I). However, the body weight of nude mice exhibited no significant differences between them (Appendix A). The immunohistochemistry results displayed a dramatic reduction in the degree of Ki67 and p-mTOR among LINC00857-silenced tumor tissues (Figure 4J and Appendix A). Our previous study showed that HSF1 stimulates mTOR activity to promote CRC progression. In this case, LINC00857 was also critical for mTOR activity, further validating the biological significance of the HSF1/LINC00857 axis in CRC.

### 3.5. The HSF1/LINC00857 Axis Promotes the Transcription of ANXA11

Previous studies have mainly referred to the function of LINC00857 in the cytoplasm. The subcellular distribution results confirmed that LINC00857 was localized in both the nucleus and the cytoplasm (Figure 5A). Therefore, our study focused on the functions of LINC00857 in the nucleus in CRC. Notably, we searched for LINC00857 gene loci using the UCSC database and found ANXA11 was at the opposite strand of the upstream region of LINC00857 where there was also a canonical HSF1 binding site (Figure 5B). The TCGA data showed a significant positive correlation between the expression of HSF1, LINC00857, and ANXA11 in CRC (Figure 5C and Appendix A). Meanwhile, we detected the increased expression of ANXA11 in CRC tissues (Appendix A) and discovered a co-expression of HSF1 and ANXA11 in CRC tissues (Figure 5D and Appendix A). The clinicopathological parameters of CRC patients were showed in Appendix A. Moreover, based on the data from the TIMER and Kaplan–Meier Plotter databases, a high ANXA11 expression was associated with a poorer prognosis (Appendix A). More importantly, the qPCR data demonstrated that the expression of ANXA11 was dramatically decreased after LINC00857 or HSF1 knockdown, respectively (Figure 5E,F). Likewise, both HSF1 and LINC00857 could stimulate the activity of the ANXA11 promoter (Figure 5G), suggesting a direct regulation of the HSF1/LINC00857 axis during ANXA11 transcription. Moreover, reintroducing ANXA11 was able to partially rescue the LINC00857-depletion-induced cell growth defect (Appendix A). Notably, our study discovered that silencing LINC00857 or HSF1 strongly attenuated the enrichment of the transcription activator RNA Pol II in the ANXA11 promoter region (Figure 5H), suggesting that the HSF1/LINC00857 axis could promote the recruitment of RNA Pol II to contribute to ANXA11 transcription. Taken together, these data indicated that LINC00857 cooperates with HSF1 to activate ANXA11 transcription in the nucleus.

### 3.6. The LINC00857/ANXA11 Axis Promotes SLC1A5-Mediated Glutamine Transport

As the above data show, the super enhancer was closely associated with cell metabolic processes. According to the lnCAR data, KEGG enrichment analysis showed that LINC00857 was the most relevant in metabolic pathways (Appendix A). In our previous study, we reported that HSF1 stimulated glutaminase 1 (GLS1)-mediated glutaminolysis to promote CRC development [22]. Therefore, we wondered whether there was a connection between the LINC00857/ANXA11 axis and glutamine metabolism. Firstly, we performed amino-acid metabolic profiling by liquid chromatography mass spectrometry (LC–MS) analysis and found that, after LINC00857 knockdown, the whole amino-acid metabolic profile was markedly altered (Figure 6A and Appendix A). Among the amino acids, glutamic acid (Glu) and aspartic acid (Asp) were significantly downregulated after LINC00857 knockdown (Figure 6B–D). Since glutamic acid can be converted to aspartic acid, it is probable that the decrease in aspartic acid is caused by the decrease in glutamic acid, validating the association between LINC00857 and glutamine metabolism. However, we did not detect the alteration of GLS1 expression after LINC00857 or ANXA11 knockdown (Appendix A). Interestingly, we identified a significant correlation between the expression of the HSF1/LINC00857/ANXA11 axis and SLC1A5 (also known as ASCT2) (Figure 6E and Appendix A), a plasma membrane transporter responsible for regulating intracellular glutamine levels [34]. In addition, we measured the decreased SLC1A5 protein levels and p-mTOR activity after LINC00857 or ANXA11 knockdown, which were in accordance with our previous results (Figure 6F and Appendix A). In summary, the LINC00857/ANXA11 axis could promote glutamine transport by regulating SLC1A5 protein expression.

However, we did not detect the alteration of SLC1A5 mRNA after ANXA11 knockdown (Appendix A), implying a novel regulatory mechanism between them. Recent studies have shown that, as with lncRNA or circRNA, mRNA can also competitively bind miRNAs to promote the translation process of downstream targets (ceRNA mechanism) or mediate miRNA degradation [35,36]. Using bioinformatics tools, we screened out a series of miRNAs complementary to ANXA11 mRNA or the SLC1A5 3′UTR region (Figure 6G,H). Among them, miR-122-5p was the only intersection which was negatively associated with the expression of ANXA11 in CRC (Appendix A). Quantitative-PCR assays showed that the knockdown of ANXA11 did not affect the expression of miR-122-5p (Appendix A), excluding the possibility that ANXA11 could mediate miR-122-5p’s degradation. Dual-luciferase assays indicated that miR-122-5p could directly bind to ANXA11 mRNA and the SLC1A5 3′UTR region, inhibiting their activity (Figure 6I and Appendix A). However, miR-122-5p failed to inhibit this activity once the binding sites were mutated in the ANXA11 mRNA and the SLC1A5 3′UTR region. Meanwhile, the overexpression of miR-122-5p significantly suppressed SLC1A5 protein expression, further demonstrating that SLC1A5 was a target of miR-122-5p. However, the inhibition of miR-122-5p could rescue SLC1A5 inactivation with ANXA11 depletion, suggesting that ANXA11 regulates SLC1A5 expression via miR-122-5p (Figure 6J,K). To determine whether ANXA11 and miR-122-5p were in the RNA-induced silencing complex, we performed RIP assays using the Ago2 antibody. We observed a higher level of ANXA11 mRNA and miR-122-5p in the anti-Ago2 group (Figure 6L,M). Taken together, these data suggest that ANXA11 could competitively bind to miR-122-5p, resulting in SLC1A5 protein expression.

### 3.7. Knockout of ANXA11 Attenuated CRC Carcinogenesis In Vivo

In our previous study [22], HSF1-null mice were highly resistant to the azoxymethane (AOM)/dextran sulfate sodium (DSS)-induced CRC mice model. Our next focus was whether ANXA11 participated in HSF1-mediated CRC carcinogenesis in the AOM/DSS-induced mice model. Firstly, we identified that ANXA11 protein levels were markedly increased in AOM/DSS-treated mice, while they were significantly decreased after HSF1 knockout (Figure 7A,B). To validate that ANXA11 indeed participated in this process, we generated ANXA11-knockout mice by a CRISPR/Cas9 strategy (Figure 7C). We observed that the number and area of tumors were noticeably reduced after ANXA11 deletion (Figure 7D–F). It is noteworthy that the overall survival of ANXA11-null mice was longer than that of the control mice (Figure 7G). Moreover, variation in body weight was not apparent (Figure 7H). Immunohistochemistry assays displayed a dramatic reduction in the degree of Ki67 among ANXA11-silenced tumor tissues (Figure 7I and Appendix A). These data suggest that ANXA11 is also indispensable for the formation of AOM/DSS-induced CRC. 

## 4. Discussion

The accumulating evidence has suggested that HSF1 exerts a multifaced feature in tumorigenesis [37]. Recent studies have demonstrated that HSF1 expression is negatively correlated with immune checkpoint genes’ expression [38], tumor mutational burden (TMB), and microsatellite instability (MSI) in CRC [39], implying no reaction to immunotherapy when HSF1 is highly expressed. Therefore, it may be theoretically feasible to target HSF1 when unresponsive to immunotherapy. Even stromal HSF1 promoted extracellular matrix remodeling, contributing to inflammation-driven CRC [40]. Given the dual effects on tumor cells and stroma, we asked whether targeting HSF1 would receive a better result. Glutamine is one of the primary nutrients on which cancer cells depend. Based on the latest studies, cancer cells prefer glutamine rather than glucose, while the tumor microenvironment consumes more glucose [41]. Therefore, it is more worthwhile to explore the regulatory mechanism of glutamine metabolism in cancer cells. Previously, we have demonstrated that HSF1 can promote CRC progression by stimulating glutamine catabolism through upregulating GLS1 expression [22]. In this study, we demonstrated that HSF1 could also stimulate glutamine metabolism by regulating the glutamine transporter-SLC1A5/ASCT2, further highlighting the potential of targeting HSF1 in colorectal cancer.

Super enhancers play key roles in driving the expression of cell-specific genes by interacting with transcription factors, RNA pol II, and non-coding RNA. Recent research has shown that the acetyltransferase P300 was responsible for the super-enhancer activity, which activated transcription by acetylating histones and promoting the binding of transcriptional complexes to target gene promoters [33,42]. However, the mechanism of how super-enhancer activity is regulated remains elusive. The “pioneer” transcription factor HSF1 could first get close to the regulatory elements and help the recruitment of other factors by acting as a scaffold. Indeed, our study found that HSF1 could stimulate the super-enhancer activity by promoting the enrichment of P300 at gene loci. Accordingly, our study shed light on the mechanism of how super-enhancer activity is regulated in CRC. Our previous study demonstrated that H3K27ac activity was indispensable for HSF1 expression [38]. Therefore, a positive feedback loop may exist between HSF1 and super enhancers, facilitating HSF1’s persistent activation.

The long chain non-coding RNA (lncRNA) directly regulated by super enhancers is called super-enhancer lncRNA (SE-lncRNA) [12]. For example, SE-LINC01503, regulated by TP63, promotes ESCC cell proliferation by activating MAPK and AKT signaling [43]. At present, the functions of nuclear SE-lncRNA are seldom explored. In our study, we identified a super-enhancer, lncRNA-LINC00857, in CRC. LINC00857 was distributed both in the nucleus and cytoplasm. Cytoplasmic lncRNA could regulate the stability or translation of transcripts, while nuclear lncRNA could be involved in transcriptional regulation by binding with TFs and DNA [44]. Previous in vitro experiments have shown that silencing LINC00857 caused dramatic defects in cell proliferation and migration through a ceRNA mechanism [45]. In accordance with their results, our study verified that LINC00857 was essential for CRC cell proliferation via in vitro and, more compelling, in vivo experiments. More importantly, we demonstrated that HSF1 and LINC00857 cooperate in the promoter region of ANXA11 and lead to its transcription. It has been previously shown that ANXA11 expression is elevated and strongly correlates with the development and metastasis of CRC [46]. These data highlight LINC00857 as a prominent super-enhancer lncRNA which could regulate ANXA11 transcription in the nucleus.

To decipher the underlying mechanism for the actions of LINC00857, we performed amino-acid metabolic profiling by LC–MS analysis. Functionally, we characterized LINC00857 as an oncogenic lncRNA by mediating glutamine metabolism. Furthermore, this effect was dependent on SLC1A5, a transporter of glutamine, rather than glutaminase. However, we did not observe changes in the level of glutamine after LINC00857 knockdown. This may because there are many sources of glutamine. Other than SLC1A5-mediated glutamine transport, glutamine could also be synthetized by glutamine synthetase by tumor cells. We consider that after LINC00857 knockdown, as with other amino acids, the level of glutamine underwent a compensatory increase so as to meet the growth needs of tumor cells. However, the compensatory ability was limited. Meanwhile, the glutaminase activity was not affected after LINC00857 knockdown. Therefore, the level of glutamate was decreased eventually. In addition, the knockdown of ANXA11 reduced the level of the SLC1A5 protein without affecting its mRNA level, implying a novel regulatory mechanism between them. Similar to lncRNA, mRNA can also competitively bind miRNAs to inhibit the translation process of downstream targets (ceRNA mechanism) [35]. In our study, we identified the regulatory mechanism between ANXA11 and SLC1A5. We found that miR-122-5p could directly bind to ANXA11 mRNA or the SLC1A5 3′UTR region, and inhibit their activity. The overexpression of miR-122-5p significantly suppressed SLC1A5 protein expression, while the inhibition of miR-122-5p could rescue the SLC1A5 inactivation with ANXA11 depletion, suggesting that ANXA11 regulates SLC1A5 expression by miR-122-5p. One study noted that circRNA_0072995 facilitates ovarian cancer progression by binding competitively to miR-122-5p, promoting the expression of SLC1A5 [47]. The mechanism by which ANXA11 modulated SLC1A5 expression was similar, forming an RNA/miRNA/RNA complex in CRC cells. Finally, our in vivo experiments also demonstrated that the genetic deficiency of ANXA11 attenuated CRC carcinogenesis, validating the results of our in vitro assays.

Here, our findings uncovered a novel mechanism of HSF1 regulation of glutamine metabolism and further emphasized the importance of targeting glutamine metabolism in colorectal cancer. Consequently, insights into the precise regulatory mechanisms of the HSF1/LINC00857/ANXA11 axis in colorectal cancer progression can further develop novel specific targeted drugs and diagnostic strategies for colorectal cancer. For example, HSF1 or ANXA11 inhibitors may be of benefit to CRC patients. Additionally, abnormal metabolism is relevant with drug resistance. Could targeting the HSF1/ANXA11 axis overcome drug resistance? Of course, the validation of these inhibitors needs to be verified by a series of in vitro and in vivo assays, providing stronger evidence for the potential clinical application of our findings.

## 5. Conclusions

Collectively, we shed light on a closely cooperative machinery between HSF1 and super enhancers which is essential for CRC malignancy (Figure 7J). HSF1 could stimulate acetyltransferase P300-mediated super-enhancer activity to facilitate LINC00857 expression, contributing to SLC1A5/ASCT2-mediated glutamine transport. In addition, targeting the HSF1/LINC00857/ANXA11 axis may provide a valuable therapeutic strategy against CRC.

## Figures and Tables

**Figure 1 cancers-14-03855-f001:**
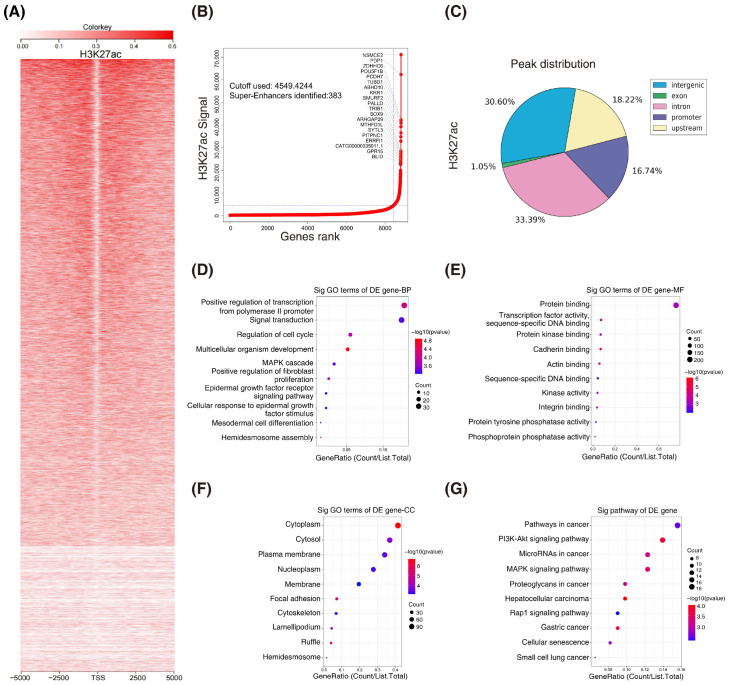
**H3K27ac landscape in colorectal cancer.** (**A**) Heatmaps of ChIP–seq signals for H3K27ac in HCT116 cells. Scale bar indicates the intensities. (**B**) Hockey stick plots on the basis of input–normalized H3K27ac signals in HCT116 cells. (**C**) Genome–wide distribution of H3K27ac ChIP–seq peaks in HCT116 cells. (**D**–**G**) Gene ontology (GO) and KEGG analyses of the super–enhancer associated genes. (BP, biological process; MF, molecular function; CC, cellular component; KEGG, Kyoto encyclopedia of genes and genomes).

**Figure 2 cancers-14-03855-f002:**
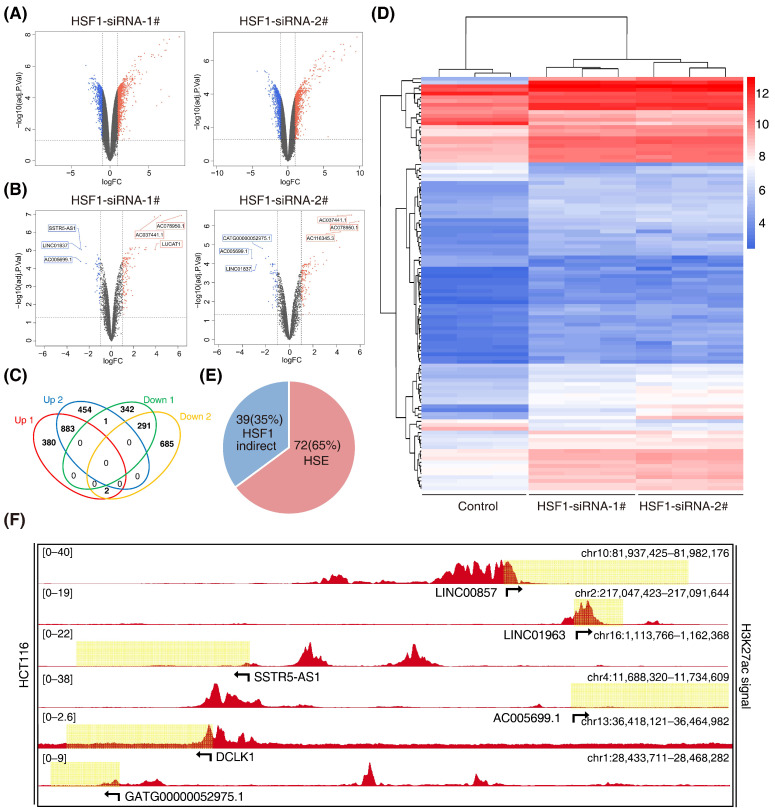
**Screening for HSF1-associated lncRNAs.** (**A**) Volcano plots of differentially expressed lncRNAs after HSF1 knockdown (*p* value < 0.05 and |log2FC| ≥ 1.0). (**B**) Volcano plots of differentially expressed enhancer-mediated lncRNAs after HSF1 knockdown (*p* value < 0.05 and |log2FC| ≥ 1.0). (**C**) Venn diagrams of overlapping differentially expressed lncRNAs from an intersection of upregulated and downregulated lncRNAs (*p* value < 0.05 and |log2FC| ≥ 1.0). (**D**) The heatmap shows the clustering of enhancer-mediated lncRNAs after HSF1 knockdown. (**E**) The number of HSF1-dependent enhancer-mediated lncRNAs with or without an HSF1 binding site (HSE) in their promoter regions. (**F**) Gene tracks of H3K27ac ChIP–seq occupancy at representative HSF1-dependent enhancer-mediated lncRNA gene loci in HCT116 cells. Data derived from the ENCODE database. Data represent the mean ± SD, *n* ≥ 3.

**Figure 3 cancers-14-03855-f003:**
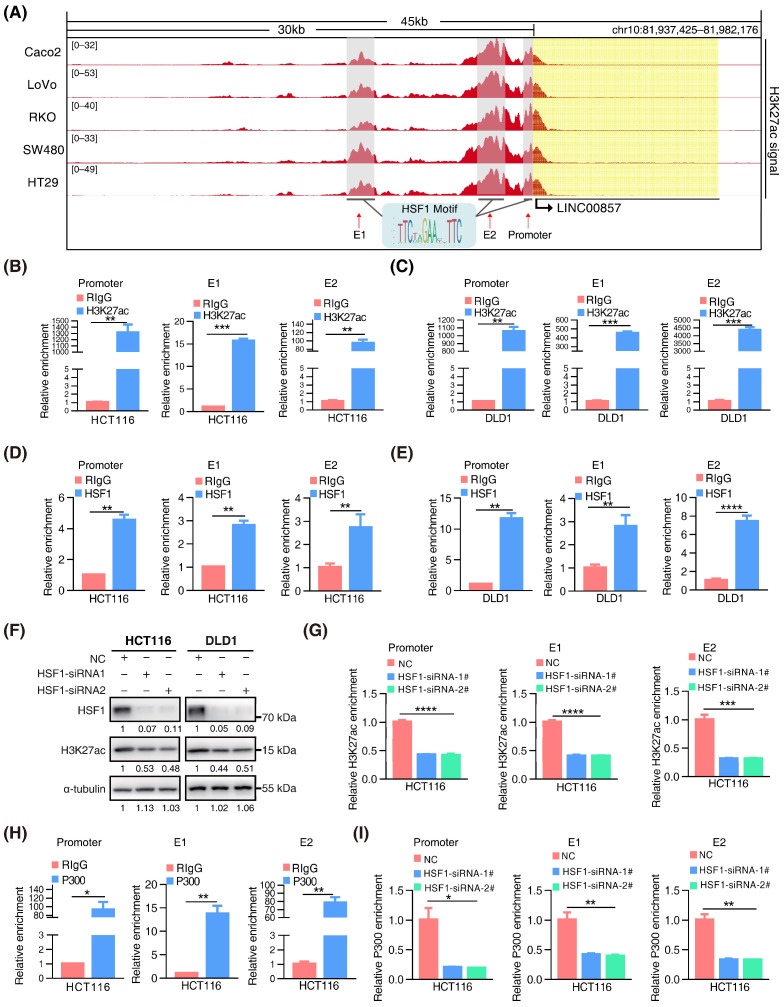
**HSF1 activates the transcription of LINC00857 via promoting its super-enhancer activity.** (**A**) Gene tracks of H3K27ac ChIP–seq occupancy at LINC00857 gene loci in CRC cell lines. Data derived from the ENCODE database. (**B**,**C**) The relative enrichment level of H3K27ac at the LINC00857 promoter and enhancers by ChIP–qPCR assays. (**D**,**E**) The relative enrichment level of HSF1 at the LINC00857 promoter and enhancers by ChIP–qPCR assays. (**F**) The effect of HSF1 knockdown on the H3K27ac level as explored by Western blotting. (**G**) The relative enrichment level of H3K27ac the LINC00857 promoter and enhancers after HSF1 knockdown by ChIP–qPCR assays. (**H**) The relative enrichment level of P300 at the LINC00857 promoter and enhancers by ChIP–qPCR assays. (**I**) The relative enrichment level of P300 at the LINC00857 promoter and enhancers after HSF1 knockdown by ChIP–qPCR assays. Data represent the mean ± SD, *n* = 3. * *p* < 0.05; ** *p* < 0.01; *** *p* < 0.001; **** *p* < 0.0001.

**Figure 4 cancers-14-03855-f004:**
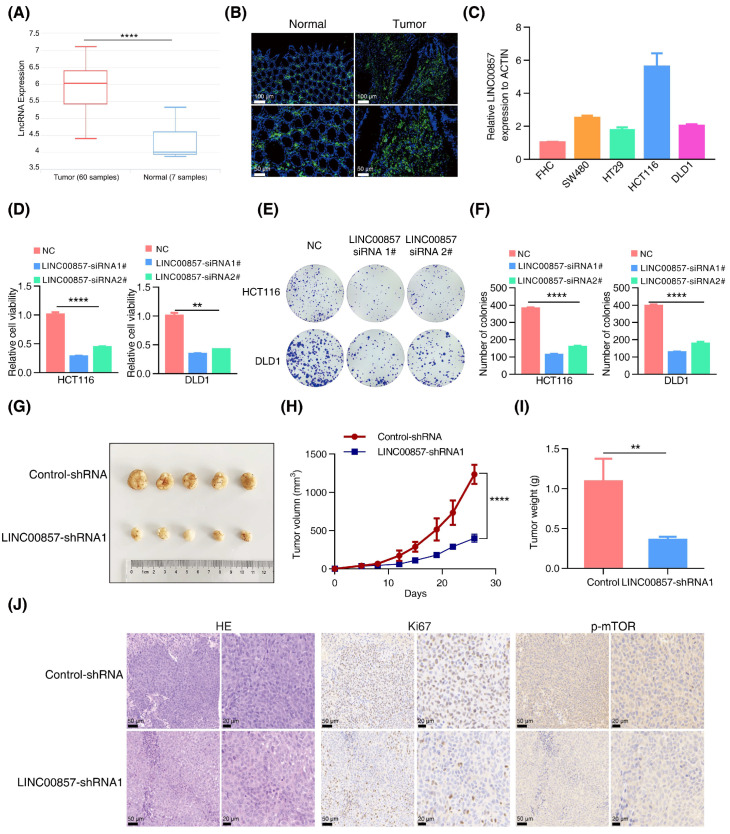
**LINC00857 is upregulated and beneficial for CRC carcinogenesis.** (**A**) Analysis of LINC00857 expression levels by data from the lnCAR database (CR_S177). (**B**) The expression of LINC00857 in CRC tissues and paired normal samples was determined by FISH (fluorescence in situ hybridization) (scale bar, 50 µm and 100 µm). (**C**) The relative expression of LINC00857 in CRC cell lines and normal colon epithelial cells (FHC) was determined by qPCR. (**D**–**F**) Relative cell viability and colony formation was measured after LINC00857 knockdown by siRNA. (**G**) Images of dissected tumors. (**H**) Tumor volumes were measured at the indicated time points in a xenograft mice model. (**I**) Summary of mean tumor weight measured at end point. (**J**) HE (hematoxylin–eosin) and immunohistochemistry analysis in scramble and LINC00857 knockdown mice (scale bar, 20 µm and 50 µm). Data represent the mean ± SD, *n* ≥ 3. ** *p* < 0.01; **** *p* < 0.0001.

**Figure 5 cancers-14-03855-f005:**
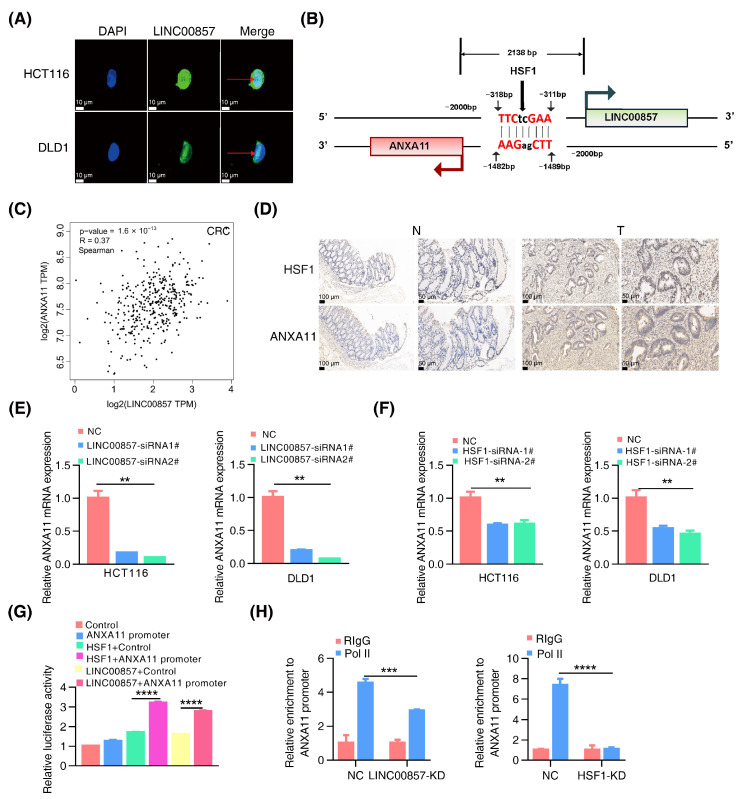
**The HSF1/LINC00857 axis promotes the transcription of ANXA11.** (**A**) Subcellular localization of LINC00857 in HCT116 and DLD1 by FISH (scale bar, 10 µm). (**B**) Diagram of LINC00857 promoter regions. The scatter plot of the relationship between HSF1, LINC00857, and ANXA11. (**C**) Correlation between LINC00857 and ANXA11 expression using the GEPIA database. (**D**) HSF1 and ANXA11 expression levels as detected in the normal and tumor tissues by IHC (immunohistochemistry) (scale bar, 50 µm and 100 µm). (**E**) Relative ANXA11 mRNA expression after LINC00857 knockdown. (**F**) Relative ANXA11 mRNA expression after HSF1 knockdown. (**G**) Relative luciferase activity after transfection HSF1, LINC00857, and ANXA11 promoter. (**H**) Relative enrichment level of RNA Pol II at the ANXA11 promoter after LINC00857 or HSF1 knockdown by ChIP–qPCR assays. All data represent the mean ± SD, *n* = 3. ** *p* < 0.01; *** *p* < 0.001; **** *p* < 0.0001.

**Figure 6 cancers-14-03855-f006:**
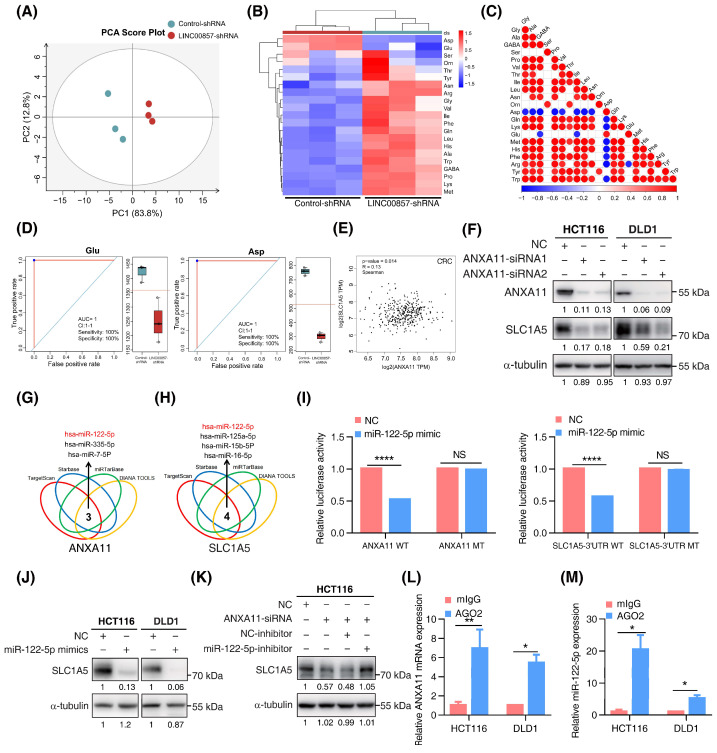
**The LINC00857/ANXA11 axis promotes SLC1A5-mediated glutamine transport.** (**A**) Score plot of principal components analysis (PCA) between control and LINC00857 knockdown group after LC–MS (liquid chromatography mass spectrometry) analysis. (**B**) Heatmap of 22 amino acids’ alteration after LINC00857 knockdown. (**C**) Association analysis between different amino acid. (**D**) Concentration of Glu (glutamic acid) and Asp (aspartic acid) after silencing LINC00857. (**E**) Correlation between ANXA11 and SLC1A5 expression using the GEPIA database. (**F**) The effect of ANXA11 knockdown on SLC1A5 expression in CRC cells, as explored by Western blotting. (**G**,**H**) Bioinformatics tools predicted a series of miRNAs complementary to ANXA11 mRNA and the SLC1A5 3′UTR region. (**I**) Dual-luciferase assays of the ANXA11 mRNA/SLC1A5 3′UTR constructions with intact or mutated seed sequences for miR-122-5p. (**J**) The effect of miR-122-5p mimics on SLC1A5 protein level, as explored by western blotting. (**K**) The effect of miR-122-5p inhibitors on ANXA11 knockdown-induced SLC1A5 downregulation, as explored by Western blotting. (**L**,**M**) RIP (RNA immunoprecipitation) assays were performed using the Ago2 antibody. The relative expression levels of ANXA11 and miR-122-5p were determined by qPCR. All data represent the mean ± SD, *n* = 3. * *p* < 0.05; ** *p* < 0.01; **** *p* < 0.0001.

**Figure 7 cancers-14-03855-f007:**
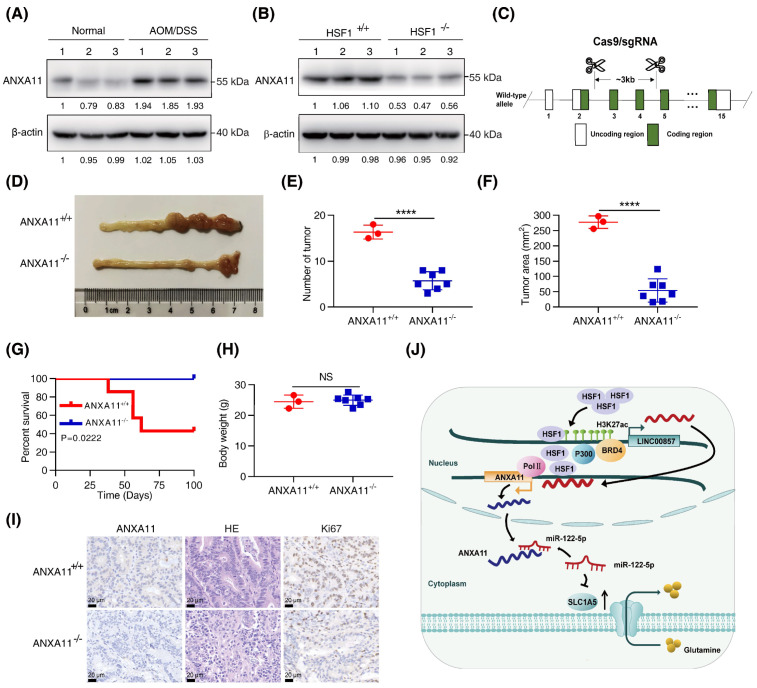
**Knockout of ANXA11 attenuated CRC carcinogenesis in vivo.** (**A**) Expression of ANXA11 protein in azoxymethane (AOM)/dextran sulfate sodium (DSS)–treated mice compared with normal mice. (**B**) Expression of ANXA11 protein in AOM/DSS–treated mice after HSF1 knockout. (**C**) Schematic digraphs depicting the CRISPR/Cas9-mediated ANXA11 knockout strategy. (**D**) Representative images of ANXA11^+/+^ and ANXA11^−/−^ mice colon and rectum. (**E**) Assessment of the number of tumors of ANXA11^+/+^ and ANXA11^−/−^ mice. (**F**) Assessment of the tumor area of ANXA11^+/+^ and ANXA11^−/−^ mice. (**G**) The overall survival of ANXA11^+/+^ and ANXA11^−/−^ mice. (**H**) The body weight of ANXA11^+/+^ and ANXA11^−/−^ mice. (**I**) Representative HE (hematoxylin–eosin) staining and IHC (immunohistochemistry) staining of Ki67 and ANXA11 from the ANXA11^+/+^ and ANXA11^−/−^ mice colorectal tissues (scale bar, 20 µm). (**J**) Working model. All data represent the mean ± SD, *n* ≥ 3. **** *p* < 0.0001.

## Data Availability

The ChIP-seq and lncRNA microarray data have been deposited to the Gene Expression Omnibus (GEO) under accession numbers GSE198223 and GSE198224.

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
