# Peer review of "HSF1 Stimulates Glutamine Transport by Super-Enhancer-Driven lncRNA LINC00857 in Colorectal Cancer"

_cancers, 2022, doi:10.3390/cancers14163855_

Round 1

Reviewer 1 Report

The study is well executed and the work supports the hypotheses. The authors present in vitro and in vivo experiments that clarify the interaction between HSF1 and super enhancers. 

I would suggest some minor improvements. The data on the patients from whom samples were collected may be included for more clarity (e.g., data on the tumor location, stage, MSI status, etc.). 

I would suggest more discussion on how the findings may be applicable to clinical practice and change practice in CRC patients.

Otherwise, the methods and results are well-presented.

Author Response

  1. We checked all the cited references again and deleted the irrelevant references.

2.The data on the patients from whom samples were collected may be included for more clarity (e.g., data on the tumor location, stage, MSI status, etc.).

Reply: We provided more patients data in the Supplemental Table 4.

  1. I would suggest more discussion on how the findings may be applicable to clinical practice and change practice in CRC patients.

Reply: We discussed the potential clinical practice in the last paragraph of the discussion section.

Reviewer 2 Report

In this article, the authors explored the function of HSF1 mediated activation of the long non-coding RNA, LINC00857. They showed that HSF1-LINC00857 positively regulate the transcription of ANXA11 which in turn modulate the level of SLC1A5 by competitively binding to miR-122-5p binding. The results are generally convincing. I have the following comments:

·      Figure 1D-G: Since the authors are interested in super enhancers, they should perform pathway analysis for 614 super enhancer associated genes as well. Also, what does DE gene in the title of figure 1D-G mean?

·      Line #218-219: It is confusing as to why the authors performed lncRNA microarray analysis to study the association of HSF1 and super-enhancers. It makes the reader think why lncRNA analysis and why not DNA motif analysis on the super enhancers associated genes to check TF binding site. Since the authors are interested in HSF1 mediated lncRNA expression, they should clearly state that.

·      Line 223: do the authors mean 65% of the enhancer-mediated lncRNAs have canonical HSF1 binding site? It should be written clearly.

·      Figure 2A: It would be informative to annotate (different color) the enhancer-mediated lncRNA in the volcano plot and label few of them.

·      In figure S2: How did the authors perform GSEA on 8 lncRNAs? Did they use GSEA specific for lncRNA like lncGSEA? If so, this should be described in the methods section.

·      The authors nicely showed that downregulation of LINC00857 had profound growth defect both in vitro and in vivo. To show the specificity to ANXA11, they should rescue the growth defect with by reintroducing ANXA11, at least in vitro.

·      Quantification for IHC should be provided. Also, it’s not clear why the authors tested the level of p-mTOR. This should be explained in the text.

·      Figure 5A: although the authors claim that LINC00857 expressed in both cytoplasm and nucleus, it seems to colocalize with DAPI alone. This should be addressed.

·      SLC1A5 imports glutamine. Why did the authors not observe changes in the level of glutamine after LINC00857 KD?

·      Figure 6: The text should be rephrased (line #347-348 and also in discussion). It’s not clear how ANXA11 binding to miR-122-5p modulate the level of SLC1A5.

Minor Comments:

·      Language needs to be revised, especially in discussion and methods section.

·      Many figures are of poor quality. The legends and labels are not legible.

·      The authors should include the cell line used for luciferase activity assay.

Author Response

  1. Figure 1D-G: Since the authors are interested in super enhancers, they should perform pathway analysis for 614 super enhancer associated genes as well. Also, what does DE gene in the title of figure 1D-G mean?

Reply: Previously, we assigned all H3K27ac enriched genes for GO and KEGG analysis. Now, we performed similar analysis for the super enhancer associated genes in new Figure 1D-G.

  1. Line #218-219: It is confusing as to why the authors performed lncRNA microarray analysis to study the association of HSF1 and super-enhancers. It makes the reader think why lncRNA analysis and why not DNA motif analysis on the super enhancers associated genes to check TF binding site. Since the authors are interested in HSF1 mediated lncRNA expression, they should clearly state that.

Reply: Because super enhancer associated mRNA have had many relevant reports. However, super enhancer associated lncRNA was seldom studied in colorectal cancer. Therefore, this study put emphasis on the HSF1-associated SE-lncRNA. We explained the reasons in the Results 3.2 section.

  1. Line 223: do the authors mean 65% of the enhancer-mediated lncRNAs have canonical HSF1 binding site? It should be written clearly.

Reply: Yes. We wrote the relevant results again in this section.

  1. Figure 2A: It would be informative to annotate (different color) the enhancer-mediated lncRNA in the volcano plot and label few of them.

Reply: We found it was not feasible to annotate (different color) the enhancer-mediated lncRNA in one volcano plot. Whether the reviewer could provide us the relevant codes? And we performed another volcano plot to annotate the enhancer-mediated lncRNA and labeled few of them in new Figure 2B.

  1. In figure S2: How did the authors perform GSEA on 8 lncRNAs? Did they use GSEA specific for lncRNA like lncGSEA? If so, this should be described in the methods section.

Reply: We first calculated the correlation between each differential lncRNA and all mRNAs, and then use GSEA preranked in GSEA software to conduct gene set enrichment analysis. We added this information to the methods 2.11 LncRNA microarray analysis section.

  1. The authors nicely showed that downregulation of LINC00857 had profound growth defect both in vitro and in vivo. To show the specificity to ANXA11, they should rescue the growth defect with by reintroducing ANXA11, at least in vitro.

Reply: We rescued the growth defect by reintroducing ANXA11 in vitro. The results were showed in new Figure S5G.

  1. Quantification for IHC should be provided. Also, it’s not clear why the authors tested the level of p-mTOR. This should be explained in the text.

Reply: We provided the quantification for IHC in new Figure S4H, S5D and S6N. And we explained the reason why we tested the level of p-mTOR in the Results 3.4 section.

  1. Figure 5A: although the authors claim that LINC00857 expressed in both cytoplasm and nucleus, it seems to colocalize with DAPI alone. This should be addressed.

Reply: The Figure 5A showed that LINC00857 was expressed in both cytoplasm and nucleus. Because the nuclei of the colorectal cancer cells were large, it seemed to colocalize with DAPI alone.

  1. SLC1A5 imports glutamine. Why did the authors not observe changes in the level of glutamine after LINC00857 KD?

Reply: There are many sources of glutamine. Other than SLC1A5-mediated glutamine transport, glutamine could also be synthetized by glutamine synthetase by tumor cells. We considered that after LINC00857 KD, like other amino acids, the level of glutamine was compensatory increased so as to meet the growth needs of tumor cells. But the compensatory ability was limited. Meanwhile, the glutaminase activity was not affected after LINC00857 KD. Therefore, the level of glutamate was decreased eventually. We added this possible reason to the discussion section.

  1. Figure 6: The text should be rephrased (line #347-348 and also in discussion). It’s not clear how ANXA11 binding to miR-122-5p modulate the level of SLC1A5.

Reply: We rephrased this part in the Results 3.6 and discussion section.

  1. Language needs to be revised, especially in discussion and methods section.

Reply: We revised the discussion and methods section and corrected some grammar errors.

  1. Many figures are of poor quality. The legends and labels are not legible.

Reply: We provided new figures with high quality in the manuscript and supplemental materials.

  1. The authors should include the cell line used for luciferase activity assay.

Reply: We added the cell line used for luciferase activity assay in the methods 2.9 luciferase activity assays section.